# Smart Superhydrophobic Filter Paper for Water/Oil Separation and Unidirectional Transportation of Liquid Droplet

**DOI:** 10.3390/membranes12121188

**Published:** 2022-11-25

**Authors:** Yu-Ping Zhang, Ning Wang, De-Liang Chen, Yuan Chen, Meng-Jun Chen, Xin-Xin Chen

**Affiliations:** 1College of Chemistry and Materials Engineering, Hunan University of Arts and Science, Changde 415000, China; 2College of Chemistry and Chemical Engineering, Henan Institute of Science and Technology, Xinxiang 453000, China; 3Changde Zhengyang Biotechnology Co., Ltd., Changde 415000, China

**Keywords:** superhydrophobic filter paper, Janus membranes, dip coating, emulation separation, nanosecond laser

## Abstract

Water/oil separation from their mixture and emulsion has been a prominent topic in fundamental research and in practical applications. In this work, a smart superhydrophobic membrane (SHP) was obtained by dipping an off-the-shelf laboratory filter paper in an ethanol suspension of trichloro (1H,1H,2H,2H-tridecafluoro-n-octyl) silane, tetraethyl orthosilicate, and titanium dioxide nanoparticles with different dimensions of 20 and 100 nm. The selection of membrane substrates was optimized including different quantitative and quantitative filter papers with different filtration velocity (slow, intermediate, and fast). The as-prepared SHP was demonstrated to be superhydrophobic and photosensitive, which was used in the separation of carbon tetrachloride and water from their mixture and emulsion. Moreover, orderly aligned micropores were formed for the modified superhydrophobic filter papers by using nanosecond laser. Unidirectional penetration was obtained for the UV-irradiated paper with a bored pore in the range of 50–500 μm in the systems of air/water and water/oil. This study may promote the understanding of unidirectional transportation of liquid droplet and facilitate the design of interfacial materials with Janus-type wettability.

## 1. Introduction

Surface wettability plays a key role in the design of special membrane materials for oil/water separation, emulsion separation, and preparation of Janus membranes (JMs) [1,2,3,4,5]. Recently, demand has increased for membranes that have intelligent responses to the environment or precisely controlled liquid unidirectional transportation. These properties are difficult to achieve using conventional fabrication techniques [6,7,8,9]. Because of its porous structure, filter paper with microfibers is widely used for adsorption of liquids and separation of solids and liquids. Filter paper could be used for treatment of oily wastewater because it is inexpensive, eco-friendly, and easily manufactured [10]. However, it is superhydrophilic, which makes its impractical in terms of selectivity. Metal mesh has extraordinary mechanical strength and usually has a large flux for separation of layered oil/water but is less effective at separating surfactant-stabilized emulsified oil because of its large openings [11,12,13]. Polymer-based membranes are highly efficient but not environmentally friendly [14,15]. Therefore, to achieve high rejection of emulsified oil, membranes with surface modifications and smaller pore sizes are required. However, reducing the pore size will decrease the flux and can require the use of external pressure. Cellulose-based papers are favored over other substrates in the design of new sophisticated membrane-type materials [16,17]. Because they are thin, lightweight, and flexible, paper-based membranes can be more readily shaped than other substrates such as silicon wafers, glass slides, metal oxides, and many polymeric materials. In addition, because of the presence of many hydroxyl groups on cellulose fibers, paper can be easily modified through covalent attachment of various chemicals. Moreover, adjustment of morphological properties such as pore size, membrane thickness, grammage, and fiber density is easy, and this enables tailoring of the paper substrate according to the requirements of the end-use product. To the best of our knowledge, there have only been a few reports of paper-based materials for oil/water separation, including Cu(OH)_2_-coated paper [18], hydrogel-coated paper [17], and superhydrophobic filter paper prepared by the dipping method [19]. However, regulation of pore size remains challenging for emulsion separation by a simple and green method.

JMs are a new concept for separation that have opposing properties on their two faces, such as hydrophilic/hydrophobic wettability or positive/negative charge [20]. Asymmetric surface wettability on the two faces is crucial for a JM, but it is not enough to realize unidirectional transport for water and oil droplets as well as bubbles. Unidirectional solute transfer at the microscopic scale remains challenging and is limited by the thickness of the hydrophobic and hydrophilic layers, the pore size, and the surface chemistry of the separation membrane. Additionally, most of the methods currently available for JM fabrication use specific instruments and are complicated, and this limits the practical application of JMs. Therefore, it is crucial to design facile strategies to fabricate JMs with excellent unidirectional transportation properties. Micro/nanosecond laser processing has attracted increasing attention because the equipment is inexpensive, the processing speed is fast, and there is no environmental pollution [21]. Laser technology is often used to produce materials for suitable microstructures. The use of laser technology to control the wettability of solid surfaces has been widely investigated but limited to metal substrates [22].

In this work, paper-based functional materials with photosensitive performance were prepared by the low cost and environment-friendly dip coating technology. The surface roughness and superhydrophobic performance of filter papers were successfully adjusted by the formula optimization and the selection of the suitable filter paper. The oil/water mixture and its emulsion were successfully separated using the resulting paper. The superhydrophobic papers were further bored by low-cost nano second laser engraving machine, followed by UV-irradiation. The as-prepared paper with enlarged pore size exhibited obvious Janus-type wettability and the possible mechanism of unidirectional penetration was proposed.

## 2. Materials and Methods

### 2.1. Materials

Qualitative and quantitative cellulose filter papers with different filtration velocity were selected as the membrane substrates. Different specifications included filter paper 201 (pore size: 20–25 μm; U = 7 cm, fast filtering speed), paper 202 (pore size: 15–20 μm; U = 7 cm, moderate filtering speed), and paper 203 (pore size: 10–15 μm; U = 7 cm, slow filtering speed), which were purchased from Hangzhou Special Paper Industry Co., Ltd. (Hangzhou, China). Trichloro(1H,1H,2H,2H-tridecafluoro-n-octyl) silane (FOTS), tetraethyl orthosilicate (TEOS), both titanium dioxide nanoparticles of 20 nm and 100 nm were purchased from Sinapharm Chemical Reagent Co., Ltd. (Shanghai, China). Carbon tetrachloride, Sudan Ⅳ, methylthionine chloride, and surfactants of Tween 80 and Span 80 were purchased from Aladdin industrial corporation (Shanghai, China). 

### 2.2. Modification and Characterization of the Filter Paper

In a 25 mL centrifuge tube, a mixture of a 10 mL ethanol suspension was prepared and mixed fully under ultrasonication for 10 s, which contained 1% (*v/v*) trichloro(1H,1H,2H,2H-tridecafluoro-n-octyl) silane (FOTS), 1% (*v/v*) tetraethyl orthosilicate (TEOS), and TiO_2_ nanoparticles with different concentration of 0.3%, 0.5%, 1.0%, and 2.0%, *w/v*, respectively. The dip-coating method was employed to modify the selected 24 kinds of filter papers by varying the percentage of TiO_2_ nanoparticles in the range of 0.3–2.0%. Note that two different size ranges of TiO_2_ nanoparticles (100 nm and 20 nm) were contained with the identical percentage in the selected ethanol suspension. Specially, a series of qualitative and quantitative filter papers with slow, medium, and fast speed were immersed into as-prepared suspension solution about 10 min, then taken out and dried at ambient temperature overnight. The surface morphologies of the original and modified filter paper were characterized by scanning electron microscope (SEM, Hitachi Su5000, Tokyo, Japan) and Energy Dispersive X-ray Spectrometer (EDS, Oxford Instruments Ultim Max, Oxford, England). The surface wettability of filter paper was evaluated by a contact angle analyzer (TST-200, Shen Zhen Testing Equipment Co., Ltd., Shenzhen, China), and the droplet size of deionized water was controlled to be 10 μL.

### 2.3. Boring the Filter Paper by Nano Second Laser

Surface laser irradiation was used to pore the filter paper by nanosecond laser engraving machine with the type KN120 (Zhengzhou Huagu Laser Co., Ltd., Zhengzhou, China). including a galvanometer scanning system to move the beam over the surface as shown in Figure 1. The filter papers were bored using a fiber pulsed laser, with a central wavelength of 1064 nm and an average power of 12 W at the repetition frequency of 20 kHz. The laser beam was delivered by a scanning mirror system in X- and Y-axes. The as-prepared samples were irradiated by a hundred-nanosecond pulsed laser in 45° crossing with the pulse width of 100 ns, the scanning speed and spacing kept 10 mm/s and 25 μm, respectively. Arrays of micropore structures were directly bored on the paper substrate. The distance between adjacent micropores was kept constant. In brief, 100 pieces of pores in the middle square area of the filter paper (2 cm × 2 cm) were bored with 10 pores for each column and each row at the x- and y-directions. Because the energy density of nanosecond laser is high far beyond the ablation, and the cellulose materials are easily removed by the extra energy.

### 2.4. Preparation of Oil/Water Separation and Surfactant-Stabilized Emulsion Separation

Carbon tetrachloride-in-water (O/W) and water-in-carbon tetrachloride (W/O) emulsions were prepared by mixing water, oil, and surfactant under magnetic stirring for 6 h. Surfactants including Tween 80 and Span 80 were used as an emulsifier with the same concentration 0.4 g/L, respectively. Surfactant-stabilized (O/W) was prepared by mixing 99% (*v/v*) of deionized water (DI), 1% carbon tetrachloride (*v/v*) and Tween 80. The preparation of W/O emulsions followed the same procedure with 99% (*v/v*) of carbon tetrachloride, 1% DI water (*v/v*), and Span 80. Both emulsions above were stable at room temperature and no obvious agglomeration or precipitation was observed overnight. An upright glass filtration device assembled by an upper glass tube and a lower glass tube with Teflon fixtures was used for the oil/water and emulsion separation. To separate the mixture of water and carbon tetrachloride, its O/W and W/O emulsions, 20.0 mL of the feed solution was poured into the upper glass tube to generate a liquid column. The whole filtration was achieved due to the contribution of the static pressure caused by gravity. The separation process was terminated when no oil filtrated through the paper membrane.

## 3. Results

### 3.1. Fabrication of the Modified Filter Paper

It is well known that a superhydrophobic surface can be fabricated based on the synergic effects of both micro/nanocomposite surface structure and low surface energy materials. Herein, a facile procedure to modify the filter papers by dip-coating method was attempted. Considering that the wettability of surface is controlled by the chemical composition and the surface roughness, we added TiO_2_ nanoparticles with two different sizes together with one or two silane couplers (TEOS and FOTS) with different alkyl lengths to increase the surface hydrophobicity and roughness of filter paper. FOTS was added as a coupler for the decrease of low surface energy on the paper surface. The filter papers were immersed in 10 mL ethanol suspension, which contained 1% FOTS, 1% TEOS, and different weights of TiO_2_ nanoparticles ranging from 0.3% to 2.0%. The excess TiO_2_ nanoparticles may cause the cracks or detachments of the coating layer and few nanoparticles cannot create enough roughness and superhydrophobicity. Herein, 24 kinds of superhydrophobic filter papers were finally obtained by immersing six kinds of filter papers in 4 kinds of suspensions with TiO_2_ nanoparticles of 0.3%, 0.5%, 1.0%, and 2.0%, respectively.

Evaluation of the wettability was attempted using water drop impact experiments, which were carried out using a Revealer high-speed camera (Fuhuang Agile device, Anhui, China) at a rate of 15,000 frames per second. The Weber number (We) is defined as We = ρRν^2^/γ, where ρ is the density, R the radius of the drop, ν is its impact velocity, and γ is the surface tension of water (Zhang et al., 2021; Lu et al., 2015). The unperturbed radius of the water droplet (10 mL) is r = 1.34 mm, and the impact velocity (ν) was 0.45 ms^−1^, corresponding to the We value of 3.72. Selected snapshots of water droplets impinging on the uncoated and coated surfaces are shown in Figure 2 at a release height of 1 cm. For the pristine qualitative paper with slow filter speed, water impinged on the superhydrophilic surface without bounce and spread utterly on the surface finally, whilst water droplet tended to bounce instead of wetting the surface after modification with 0.3% TiO_2_ nanoparticles. The water droplet finally seated on the filter paper with a spheroidal shape, indicating its superhydrophobic property.

In principle, all the modified filter papers were superhydrophobic, which could be used for oil/water separation. When the layered mixture of carbon tetrachloride (10 mL) and water (10 mL) was poured into upper vessel, the lower layer of oil (carbon tetrachloride) rapidly permeated through the modified filter paper while the upper layer of blue dyed water with methylene blue was retained in Figure 3a. Similar test based on the above paper was also illustrated using a mini separation experiment in Figure 3b. A mixture of red-dyed hexane droplet and blue-dyed water droplet was dropped on the modified filter paper. Hexane droplet penetrated the filter paper totally but water droplet was blocked on the paper surface.

Usually, the membrane with superhydrophobicity in air and superoleophilicity underwater is beneficial to the separation of W/O emulsion. Herein, all modified filter papers were identically superhydrophobic, which could be used for the oil/water separation, but only the W/O emulsion was successfully carried out by using the qualitative paper with slow filtration velocity, which were immersed in an ethanol suspension with TiO_2_ nanoparticles (2%, *w/v*). It was probably attributed that the filter paper possessed the least pore size (10–15 μm) and more TiO_2_ nanoparticles were attached into the cellulose framework of the filter paper. Furthermore, the membrane with the property of superhydrophilicity in air and superoleophobicity underwater is generally desirable to the separation of O/W emulsion, the prepared O/W emulsion was thus difficult to be separated by any modified filter paper. As shown in Figure 4a, the as-prepared qualitative paper was characterized by SEM at different magnitude and it exhibited the superhydrophobicity with a WCA of 158°. As indicated by the arrows, many nanoparticle aggregates were tightly attached to the stem of cellulose on the surface of filter paper. The chemical composition of the modified filter paper was confirmed by EDS. It can be clearly observed that the surface of the modified paper contains many elements including C, O, F, Si, Cl, and Ti. EDS analysis in Figure 4b exhibited that some main peaks corresponding to six elements were observed with different weight and element percentages (see the inlet table in Figure 4b). The superhydrophobic filter paper could separate non surfactant oil–water mixture. However, for practical application, it is important to separate surfactant-stabilized emulsion. It is worth noting that emulsion separation of the coated filter papers is much more complex than the oil/water separation behavior. Although all the coated filter papers in our experiments could be used for the oil/water separation, the surfactant-stabilized emulsified solution (oil/water and water/oil) is more difficult to separate because of its small liquid droplet size and good stability. W/O emulsion is formed by a mixture of oil, water and Span 80 surfactant as an emulsifier. The form of the system is with the water dispersed in the oil as small droplets. The aqueous phase is an internal or dispersed phase, and the oil is an external phase or dispersed medium. Whilst the oil is an internal or dispersed phase, and the aqueous phase is an external phase or dispersed medium for O/W emulsion, which is formed by a mixture of water, oil, and Tween 80 surfactant as an emulsifier. For emulsion separation, the effective pore size of membrane should always be smaller than the size of dispersed internal droplets. The incomplete separation is ascribed to the fact that the pore size of the paper is relatively large so that a few tiny droplets can penetrate through the paper. In the process of water-in-oil emulsion separation, water droplets were blocked on the modified paper, and the droplet volume increased upon coalescence with other water droplets in the oil phase. It was observed that the milky water-in-carbon tetrachloride emulsion turned from opaque (left) to transparent (right) after permeating through the modified filter paper, as shown in Figure 5a. Moreover, the optical microscope images before (left) and after (right) separation were shown in Figure 5b, respectively. Densely packed water droplets were detected in the feed, while no water droplets were observed in the filtrate, demonstrating the successful removal of tiny water droplets by the coated filter paper. The fabrication method is facile, environmentally friendly, and practical for both layered oil/water and W/O emulsion separation.

### 3.2. Constructing Water-Unidirectional Janus Membrane

The surface of the as-prepared superhydrophobic paper could be transformed to superhydrophilic one after UV irradiation 150 min according to our previous study, but the unirradiated face was still superhydrophobic [23]. During UV irradiation, photosensitive TiO_2_ nanoparticles on the paper surface generated holes that readily reacted with lattice oxygen, which led to surface oxygen vacancies. The resulting vacancies coordinated to water molecules, which caused the transformation of surface wettability from superhydrophobic to superhydrophilic. In this case, both sides of the filter paper remained opposite surface wettability, but unidirectional penetration of water droplet did not occur due to its small pore size and thick superhydrophobic thickness of the modified paper. Longer time of UV irradiation might increase the thickness of superhydrophilic layer, which benefited the unidirectional transportation of water droplet [24]. Unfortunately, the unidirectional penetration of water droplet was still not observed after 24 h UV irradiation (254 nm). It was speculated that the dense structure with smaller pore size of the modified paper strongly inhibited the penetration of UV light. Herein, a simple method to bore the filter paper via the nanosecond laser was attempted for the first time, which provided uniform circular apertures with different diameters in the range of 50–500 μm. The surface wettability of the superhydrophobic papers (modified by 0.3% TiO_2_) and their bored papers with a pore size from 50 μm to 500 μm were measured, all bored papers still remained superhydrophobic with the WCAs larger than 150° in Figure 6a. Some typical papers with the pore size of 50 μm, 250 μm, and 500 μm were characterized by SEM in Figure 6b–e and the properties of the Janus type paper were thus investigated. The pristine paper without dipping modification exhibited a coarse network with relatively large pore size (Figure 6b). After immersion in a suitable formula suspension, more nanoparticles were attached on the fibers tightly, which formed a denser membrane, so that the roughness was increased and the pore size of the paper was thus reduced (Figure 6b–e). The results demonstrated that although the functionalized paper samples possessed an apparent change of surface wettability after UV irradiation and exhibited a dramatic wetting contrast between the two sides, only the bored papers exhibited the unidirectional transportation of liquid droplet. It was attributed that UV irradiation could lead to the faster transformation of surface wettability. After the superhydrophobic papers were bored with different pore size, UV could easily penetrate the pores, which caused a gradual change quickly from superhydrophobic to superhydrophilic thickness.

The resultant Janus papers could exhibit directional water droplet gating behavior in the air-water system, which are illustrated in Figure 7a. Additionally, in oil-water systems, the Janus papers showed directional gating of droplets with integrated selectivity for either oil or water. For example, a modified paper with a bored pore size of 100 μm was used for the investigation of unidirectional penetration after 1 h UV irradiation. As shown in Figure 7b, a single water droplet (25 L) was blocked from the hydrophobic side to the hydrophilic side in the air–water system. After the UV irradiation time was increased to 1.5 h, unidirectional penetration of the water droplet occurred. Water droplet can penetrate spontaneously from the hydrophobic side to the hydrophilic side (positive direction), while the water droplet is blocked and spreads on the hydrophilic side when the Janus paper is turned over (negative direction) in Figure 7c. Similar phenomena were observed for the modified papers with different pore size in the range of 100–500 μm. For example, the transport and blockage behaviors were observed from the side view using a bored paper with a pore size of 300 μm in Figure 7d. Water could infiltrate through the paper from superhydrophobic to superhydrophilic layer but blocked from superhydrophilic to superhydrophobic layer. It should be noted that less UV irradiation time was needed with the increase of bored pore diameter. Unidirectional water droplet penetration across the Janus paper was attributed to its anisotropic critical breakthrough pressure (*P_C_*) [25,26,27]. Janus paper with heterogeneous surface wettability was illustrated with a larger *P_C_* for water at the supehydrophilic surface than at the superhydrophobic surface. When a water droplet was placed on the superhydrophobic side, it experienced downward hydrostatic pressure from the force of gravity and an upward hydrophobic force. When the water droplet partly infiltrated and reached the superhydrophilic layer, the emergent capillary force pushed the water droplet downward. In contrast, when the superhydrophilic layer face upwards, the water droplet dropped onto the superhydrophilic layer was easily absorbed owing to the capillary effect, and the resultant negligible force (*P*_C_) can be offset by the superhydrophobic layer.

Unidirectional water and oil droplet penetration was investigated in a water–oil system. Liquid gating at the oil–water interface was carried out (Figure 8 and Figure 9). The JM floated on the oil–water interface without any support when it was positively or negatively aligned at the oil–water interface (Figure 8a,b). With a thin oil film adsorbed on the upper superhydrophobic layer, a water droplet contacting the superhydrophobic side exerted a large Laplace pressure (P_L_), creating a large driving force for penetration. Water droplets with a large P_L_ could break through the upper layer and reach the below superhydrophilic layer, which would pull the liquid of the same polarity across the membrane. In contrast, when the JM was negatively aligned at the oil–water interface, a water droplet contacting the superhydrophilic layer tended to spread, giving a negligible P_L_ close to zero. This limited the driving force for penetration (Figure 8c,d). The unidirectional transport of an oil droplet is illustrated in Figure 9a–d. Carbon tetrachloride, which has a higher density than water, was selected as the oil phase for a facile demonstration of this phenomenon. Consequently, the oil droplet was blocked by the positively aligned JM and transported by the reverse-aligned paper at the oil/water interface. The directional water and oil penetration across the resultant JM can be attributed to its anisotropic Pc. The ability to gate water and oil droplet transport in a unidirectional manner represents an important form of liquid manipulation and has tremendous application potential in fields involving intelligent liquid management.

## 4. Conclusions

In the present study, we demonstrated an easy and facile method for the preparation of smart superhydrophobic substrates from off-the-shelf filter paper via controlled silanization reactions. Physical and covalent attachment of both photosensitive nanoparticles and silylating reagents to the laboratory filter paper was achieved, enabling a hierarchical micro/nano surface topography and low surface energy. The resultant filter papers were successfully used for oil/water separation and W/O emulation separation was only achieved using the qualitative filter paper with slow filtration velocity and optimal nanoparticle composition. With the help of UV irradiation and the enlarged pore size on the filter paper by nano-second laser engraving machine, a Janus type paper was also fabricated with typical unidirectional liquid droplet transportation in the air-water and water-oil systems. The low-cost, eco-friendly, and easily manufactured paper-based separation materials can potentially be used for practical applications such as oil/water separation, surfactant-stabilized emulation separation, and unidirectional transformation of liquid droplet.

## Figures and Tables

**Figure 1 membranes-12-01188-f001:**
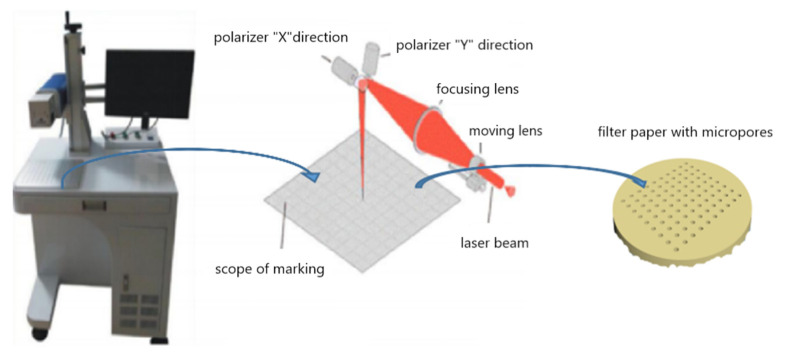
Schematic of optical setup for the fabrication of micropore on the filter paper using the nanosecond laser engraving machine.

**Figure 2 membranes-12-01188-f002:**
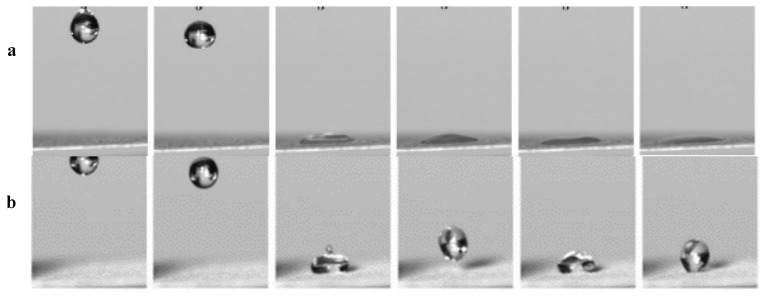
Time-lapse photographs of water droplets bouncing on the pristine qualitative filter paper (**a**) and the modified filter paper fabricated by 0.3% both nanoparticles (**b**).

**Figure 3 membranes-12-01188-f003:**
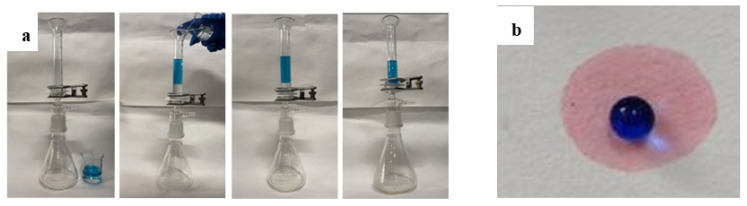
(**a**) Oil/water separation test using 20 mL mixture of carbon tetrachloride and water, and (**b**) mini-separation of a droplet of red-dyed carbon tetrachloride and a drop of blue-dyed water using the modified qualitative paper with slow filtering speed.

**Figure 4 membranes-12-01188-f004:**
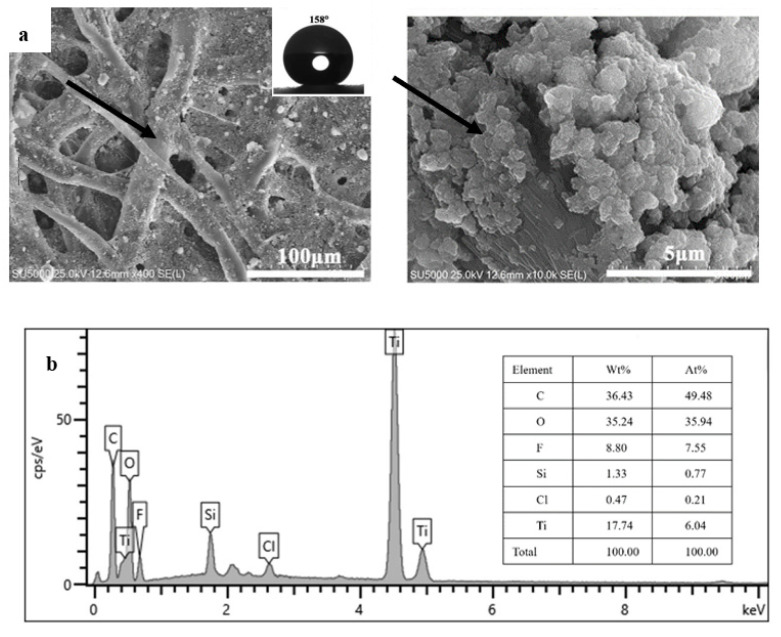
SEM images of the modified filter paper at different magnitude, inlet is the CA photo (**a**); EDS analysis of the modified filter paper (**b**).

**Figure 5 membranes-12-01188-f005:**
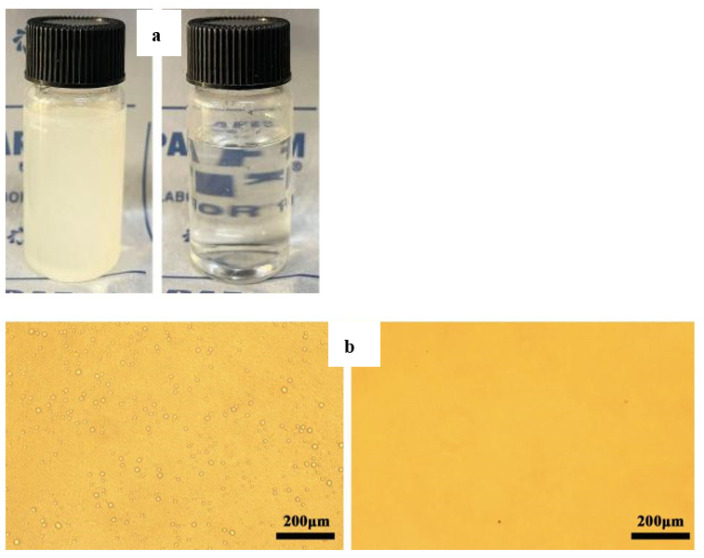
Images of water-in-CCl_4_ micro-sized emulsions before (left) and after (right) permeation through the modified filter paper. Photos of emulsion feed and filtrate (**a**), the optical microscopy images of feed and filtrate (**b**).

**Figure 6 membranes-12-01188-f006:**
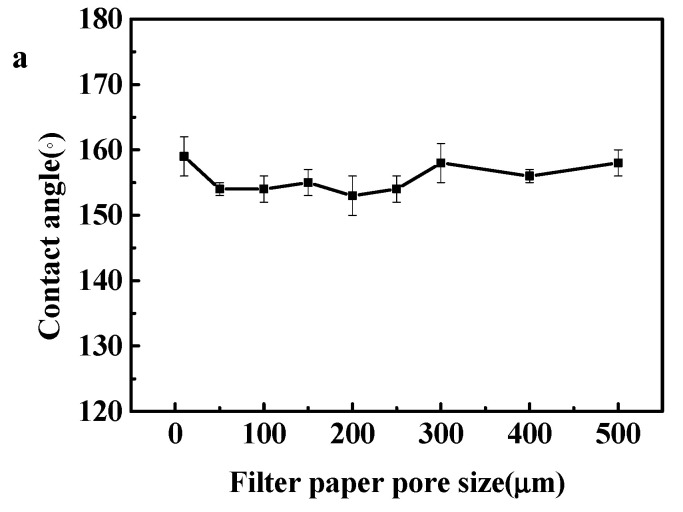
The surface wettability of the modified papers with different bored pore size in the range of 0–500 μm (**a**); SEM of some typical bored papers with different bored pore sizes of 0 μm (**b**), 50 μm (**c**), 250 μm (**d**), and 500 μm (**e**), each paper modified by 0.3% TiO_2_ nanoparticles was characterized with different magnification, respectively.

**Figure 7 membranes-12-01188-f007:**
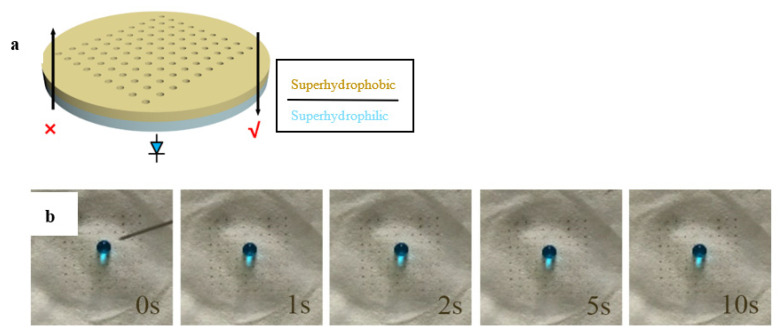
(**a**) Schematic of the the laser-treated paper samples. (**b**) Still frames taken from a video to show the dropping of blue-dyed water onto the UV-irradiated “upper” surfaces for the Janus paper (100 μm) after UV irradiation about 1 h. (**c**) Still frames taken from a video to show the dropping of blue-dyed water onto the UV-irradiated “upper” (**c1**) and “below” (**c2**) surfaces for the Janus paper (100 μm) after UV irradiation about 1.5 h. (**d**) Still frames taken from a video to show the dropping of blue-dyed water onto the UV-irradiated “upper” (**d1**) and “below” (**d2**) surfaces for the Janus paper (300 μm) after UV irradiation about 1 h.

**Figure 8 membranes-12-01188-f008:**
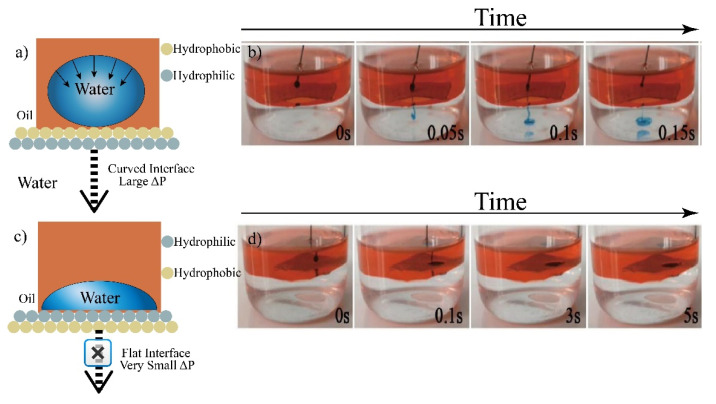
Unidirectional water droplet penetration across Janus paper (300 μm) in the oil–water system. (**a**,**b**) For positively aligned Janus paper, a water droplet touching the superhydrophobic side exerts a larger Laplace pressure (P_L_), creating a larger driving force for penetration. (**c**,**d**) For reversely aligned Janus paper, a water droplet seated on the superhydrophilic side tends to spread, exerting a smaller P_L_; therefore, the oil-infused layer can block its penetration.

**Figure 9 membranes-12-01188-f009:**
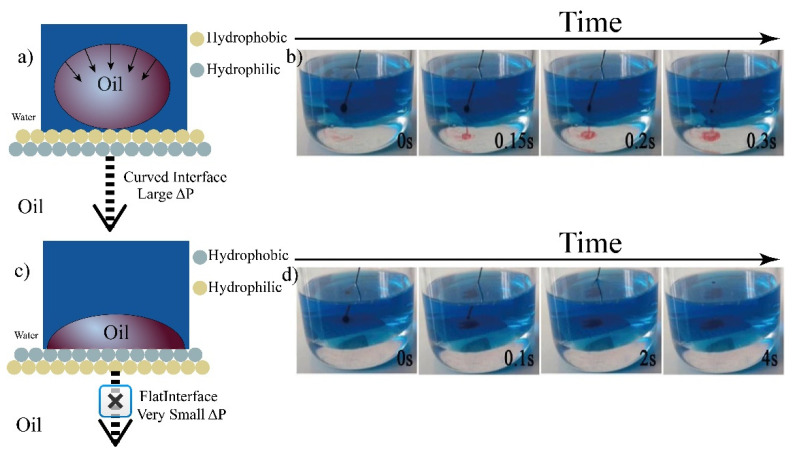
Unidirectional oil droplet penetration across Janus paper (300 μm) in the water–oil system. (**a**,**b**) For negatively aligned Janus paper, an oil droplet (carbon tetrachloride) touching the superhydrophilic (oleophobic underwater) side exerts a larger P_L_, creating a larger driving force for penetration. (**c**,**d**) For positively aligned Janus paper, an oil droplet seated on the hydrophobic (oleophilic underwater) side tends to spread, exerting a smaller P_L_.

## Data Availability

Data are contained within the article.

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
