# Peer review of "Smart Superhydrophobic Filter Paper for Water/Oil Separation and Unidirectional Transportation of Liquid Droplet"

_membranes, 2022, doi:10.3390/membranes12121188_

Round 1
Reviewer 1 Report
Smart hydrophobic membrane for oil water separation was prepared by dipping an off-the-shelf laboratory filter paper in an ethanol suspension of trichloro (1H,1H,2H,2H-tridecafluoro-n-octyl) silane, tetraethyl orthosilicate, titanium dioxide nanoparticles with different dimensions of 20 and 100 nm.
The following literature should be included in the revised version of the manuscript.
a. https://doi.org/10.1016/j.jwpe.2021.102293
b. https://doi.org/10.1016/j.memsci.2011.03.011
c. https://doi.org/10.1016/j.seppur.2021.118581
d. https://doi.org/10.1021/acs.langmuir.1c02046
e. DOI: 10.1016/j.cherd.2018.12.007
f. https://doi.org/10.1016/j.memsci.2013.06.053
g. https://doi.org/10.1016/j.apsusc.2018.04.180
h. DOI: 10.1016/j.desal.2020.114428
i. https://doi.org/10.1016/j.watres.2022.118270
Author Response
Many thanks for the reviewer’s recommended papers ! we have cited some important and highly relative papers in our revised manuscript. Also, all references involved were again adjusted for the changes.
Reviewer 2 Report
The manuscript is devoted to the separation of direct and reverse emulsions based on oil and water with paper filters, including those with a modified surface. After a series of manipulations, the authors obtain a membrane with a hydrophilic and hydrophobic surface. The question immediately arises about the limits of applicability of this method, its efficiency and economy. It is necessary to compare with the evaporation of water from the system when it is heated. It is not entirely clear how paper changes when exposed to UV. What stream values ​​can authors specify? Those. it is necessary to designate the performance of the membrane. The purpose of the work is not clearly defined in the theoretical part. Also, the theoretical part of the manuscript contains a small coverage of already published works. Line 17. "filtering speed" is an unfortunate term in my opinion, maybe it's better to use "separation"?! keywords. Replace "Emulation separation" with "separation". Line 50. "pore thickness" is an unfortunate term. Line 212. The authors observe a C content of 52.89%. However, the theoretical carbon content of cellulose is 44.4% (https://doi.org/10.3390/polym13040537 ). In my opinion, this should be paid attention to and discussed in the manuscript! Line 235. "b" - delete! Line 285. "0μm" - delete! Figures 8 and 9. The quality of the captions in the figure needs to be improved. Figure 9. Need to check the symbols!Author Response
The question immediately arises about the limits of applicability of this method, its efficiency and economy. It is necessary to compare with the evaporation of water from the system when it is heated.
Answer: Many thanks for the kind suggestion! Actually, our fabricated method is facile, friendly-environmental and practical, it can be used for both layered oil/water and emulsion separation.
Also, the used laser instrument is not expensive.
It is not entirely clear how paper changes when exposed to UV. What stream values ​​can authors specify? Those. it is necessary to designate the performance of the membrane.
The purpose of the work is not clearly defined in the theoretical part. Also, the theoretical part of the manuscript contains a small coverage of already published works.
Answer: The mechanism about the transformation of surface wettability is explained as follows: During UV irradiation, photosensitive TiO2 nanoparticles attached on the filter paper generated holes that readily reacted with lattice oxygen, which led to surface oxygen vacancies. The resulted vacancies coordinated to water molecules, which caused the transformation of surface wettability from superhydrophobic to superhydrophilic. Many previous studies focused on the fabrication of superhydrophobic filter papers with the help of surface roughness and low surface energy material, but most of papers were only used for layered oil/water separation. Herein, we adjusted the paper’s pore size for emulation separation by the optimal content of TiO2 nanoparticles. Its Janus property was further investigated after UV irradiation. We will enlarge its practical and theoretical application in our next research.
Line 17. "filtering speed" is an unfortunate term in my opinion, maybe it's better to use "separation"?! keywords. Replace "Emulation separation" with "separation". Line 50. "pore thickness" is an unfortunate term.
Answer: we have changed "filtering speed" to filtration velocity. but "Emulation separation" is still retained according to the common academic term. Line 50. "pore thickness" is revised to pore size and membrane thickness.
Line 212. The authors observe a C content of 52.89%. However, the theoretical carbon content of cellulose is 44.4% (https://doi.org/10.3390/polym13040537 ). In my opinion, this should be paid attention to and discussed in the manuscript!
Answer: The theoretical carbon content of cellulose is 44.4%. Our modified filter paper is 52.89%, due to the addition of 1% (v/v) trichloro(1H,1H,2H,2H-tridecafluoro-n-octyl) silane (FOTS) and 1% (v/v) tetraethyl orthosilicate (TEOS) in the modification suspension, some reactions occur between the -OH of filter paper and FOTS/ TEOS, which led to the increasement of carbon content.
Line 235. "b" - delete! Line 285. "0μm" - delete!
Figures 8 and 9. The quality of the captions in the figure needs to be improved.
Figure 9. Need to check the symbols!
Answer: we have deleted “b” and in line 285, "0μm" is deleted now. The quality of Fig.8 and Fig.9 is improved now.
Reviewer 3 Report
Comments to the Authors
In this manuscript authors prepared a smart superhydrophobic membrane (SHP) by dipping an off-the-shelf laboratory filter paper in an ethanol suspension of trichloro (1H,1H,2H,2H-tridecafluoro-n-octyl) silane, tetraethyl orthosilicate, titanium dioxide nanoparticles with different dimensions of 20 and 100 nm. The oil/water mixture and its emulsion were separated using the superhydrophobic membrane. This research has value for the researchers in the related areas. However, the paper needs improvement before acceptance for publication. My detailed comments are as follow:
1. In the manuscript authors should introduced following relevant articles of oil/water separation based membranes in the introduction section
a. doi.org/10.1021/acs.iecr.0c03069
b. doi.org/10.1002/app.48677
2. The nanoparticles should be marked in electron microscopic images.
3. The quality of figure 3a should be improved like other images.
4. Authors should compare its results with other studies.
5. There are few typos error. Authors should correct it in the revised manuscript
Author Response
- In the manuscript authors should introduced following relevant articles of oil/water separation based membranes in the introduction section
- doi.org/10.1021/acs.iecr.0c03069
- doi.org/10.1002/app.48677
Answer: We have cited both articles recommended according to the suggestion from the reviewer.
- The nanoparticles should be marked in electron microscopic images.
Answer: we have used the arrows to mark the nanoparticle aggregates (see Fig.4a) in our revised manuscript.
- The quality of figure 3a should be improved like other images.
Answer: we have arranged Fig.3a, now it is clear to observe the oil/water separation. With the separation proceeding, the liquid level was apparently descended.
- Authors should compare its results with other studies.
Answer: we have added some discussion about the advantage for the fabricated filter paper in our revised manuscript (see line 239). The fabrication method is facile, environmental and practical for both layered oil/water and W/O emulsion separation.
- There are few typos error. Authors should correct it in the revised manuscript
Answer: we have carefully checked the whole manuscript and corrected some typos errors.
Round 2
Reviewer 2 Report
The authors answered the questions. Unfortunately, I still have doubts about the possibility of large-scale use of these materials. I also recommend that the authors once again check the carbon and oxygen content for the materials under study.
Author Response
In a published paper titled “Robust self-cleaning surfaces that function when exposed to either air or oil.” (Science, 2015,347,1132–1135. doi:10.1126/science.aaa0946 ).
The authors have created an ethanolic suspension of perfluorosilane-coated titanium dioxide nanoparticles to create a self-cleaning surface that functions even upon emersion in oil. The paint could be sprayed, dipped on the substrates including steel, glass, cotton and paper, respectively. The paper focuses on the fabrication of superhydrophobic surfaces on different substrates using a general suspension. In our case, we used similar suspension to modify the filter paper and extend its application for layered oil/water and emulsion separation, so large-scale use of these materials is possible.
EDS provides the sample surface of qualitative or semi-quantitative component element analysis. We also characterized the modified filter paper according to the kind advice, new EDS was added in our revised manuscript now. Six elements were measured in the revised Fig.4. Many thanks for the good suggestion one more !